# On the Properties of Styrene–Maleic Acid Copolymer–Lipid Nanoparticles: A Solution NMR Perspective

**DOI:** 10.3390/polym16213009

**Published:** 2024-10-26

**Authors:** Vladislav V. Motov, Erik F. Kot, Svetlana O. Kislova, Eduard V. Bocharov, Alexander S. Arseniev, Ivan A. Boldyrev, Sergey A. Goncharuk, Konstantin S. Mineev

**Affiliations:** 1Shemyakin-Ovchinnikov Institute of Bioorganic Chemistry RAS, 117997 Moscow, Russia; motov.vladislave@gmail.com (V.V.M.); erick.kot@gmail.com (E.F.K.); bon@nmr.ru (E.V.B.); aars@nmr.ru (A.S.A.); 2Moscow Center for Advanced Studies, Kulakova Str. 20, 140829 Moscow, Russia; 3Faculty of Biology, Shenzhen MSU-BIT University, Shenzhen 518172, China; 4Frumkin Institute of Physical Chemistry and Electrochemistry, Russian Academy of Sciences, Leninsky Prospect 31, 119071 Moscow, Russia; s.o.kislova@phyche.ac.ru (S.O.K.); i_boldyrev@mail.ru (I.A.B.)

**Keywords:** membrane proteins, membrane mimetics, NMR spectroscopy, SMALP

## Abstract

The production of functionally active membrane proteins (MPs) in an adequate membrane environment is a key step in structural biology. Polymer–lipid particles based on styrene and maleic acid (SMA) represent a promising type of membrane mimic, as they can extract properly folded MPs directly from their native lipid environment. However, the original SMA polymer is sensitive to acidic pH levels, which has led to the development of several modifications: SMA-EA, SMA-QA, and others. Here, we introduce a novel SMA derivative with a negatively charged taurine moiety, SMA-tau, and investigate the formation and characteristics of lipid–SMA-EA and lipid–SMA-tau membrane-mimicking particles. Our findings demonstrate that both polymers can form nanodiscs with a patch of lipid bilayer that can undergo phase transitions at temperatures close to those of the lipid bilayer membranes. Finally, we discuss the potential applications of these SMAs for NMR spectroscopy.

## 1. Introduction

Membrane proteins attract the most interest in structural biology, as they may serve as drug targets for a wide range of severe diseases. However, studies on membrane proteins are hampered due to the need to place these proteins into an environment closely resembling the cell membrane. None of the existing high-resolution approaches can work efficiently in the presence of liposomes, the most adequate membrane mimetic. Therefore, it is necessary to utilize lipid-containing nanoparticles of relatively small size (a radius of 2–6 nm is desirable). Several types of such membrane mimetics have previously been suggested for structural studies, namely, detergent micelles, bicelles, amphipols, and lipid–protein nanodiscs (LPNs) [1,2,3,4,5,6,7,8,9,10,11,12], which all have their advantages and drawbacks. Among the most recent advances in the field, one can highlight so-called SMA–lipid particles, SMALPs, or lipodiscs [13]. 

SMA is an amphipathic copolymer of styrene and maleic acid. When added to liposomes or cell membranes, SMA can extract lipids and form discoidal particles, which are believed to contain a patch of lipid bilayer [14]. This is an essential advantage of SMA in comparison to bicelles and LPNs, because SMA can extract the properly folded proteins directly in their native lipid environment, and no detergent is used [14,15,16,17,18,19,20,21,22]. At the same time, SMALPs represent an “open system”, similar to bicelles, and SMALPs can exchange their contents upon collision [23]. Additionally, SMA can form nanodiscs of various sizes in the range of 10–60 nm, and the largest particles can align spontaneously in a strong magnetic field, which is beneficial for solid-state NMR applications [24,25]. The original SMA polymer cannot tolerate an acidic pH or the presence of divalent cations [14,26]; therefore, several modifications to SMA have been introduced (SMA-QA, SMA-EA, SMA-ED, and many others [27,28,29]). SMA nanodiscs have been successfully applied in structural studies using cryo-electron microscopy [4,30,31,32], and their principal applicability for the NMR investigation of membrane-anchored soluble domains has been demonstrated [16,33,34]. On the other hand, the possibility of the observation of transmembrane domains has not yet been shown. 

In addition to the general applicability of lipodiscs for structural studies, an essential question is whether the state of lipids in particles resembles the interior of the cell membrane. The same question stands for other membrane mimetics, including lipid–protein nanodiscs [35] and phospholipid bicelles [36,37]. One of the major properties of lipid bilayers is that they undergo phase transitions, and in the case of SMALPs, this property was investigated for particles formed by the original SMA polymer [38]. It appeared that polymers with different ratios between styrene and maleic acid affected lipid behavior to various extents, and the effect was additionally dependent on the contents of the polymer and lipids [26,27,34]. However, the properties of lipodiscs formed by the modified SMA analogs were not studied in much detail. Here, we utilize the NMR-based approach that we previously proposed to investigate the structure of lipid bicelles [36,37,39] and study the phase transitions, size, and applicability limits of lipodiscs based on the modified SMA polymers.

## 2. Materials and Methods

### 2.1. SMA-EA and SMA-Tau Synthesis

The SMA modifications used in this work were synthesized using previously characterized commercially available SMAnh (Sigma-Aldrich, Burlington, MA, USA, Mw = 1900 g/mol, ratio of styrene to maleic units of 3:1) [40].

To synthesize SMA-EA, SMAnh (1 g, 2.55 mmol) was dissolved in 50 mL of Dimethylformamide (DMF) with stirring. Ethanolamine (311.5 mg, 5.1 mmol) was dissolved in 2 mL of DMF. The solutions were mixed and stirred with magnetic stirrer (5drops-HS-19, 5drops.ru, Moscow, Russia) for 1 h (until the reaction mixture turned light yellow). A total of 50 mL CHCl_3_ was added (until the reaction mixture became cloudy and heated). Then, 30 mL of petroleum ether was added and left to stir for 2 days, during which a white crystalline precipitate formed. It was filtered under vacuum conditions and freeze-dried (INEJ-6, Institute for Biological Instrumentation of RAS, Pushchino, Russia). The yield was 30% [41].

To synthesize SMA-tau, 15 mL of distilled DMF was added to SMAnh (300 mg, 0.765 mmol) and stirred for an hour. Taurine (191 mg, 1.53 mmol) and KOH (86 mg, 1.53 mmol) were dissolved in 5 mL distilled water. The solutions were drained and stirred for two hours. The resulting solution was evaporated on a rotary evaporator (Rotavapor R-300, Buchi, Flawil, Switzerland). The resulting white crystals were lyophilized. Then, they were dissolved in 10 mL of water, and dialysis was carried out (the dialysis bag used allows substances weighing up to 1 kDa to pass through). The bag was placed in a 1 L glass container and stirred overnight. The yield was 58%.

### 2.2. Protein Expression and Purification

p75-TM-ICD was expressed in *E. coli* strain BL21(DE3) in M9 medium. To obtain the ^15^N-labeled protein, ^15^NH_4_Cl was used. The cells were grown overnight at 28 °C in a shaking incubator (New Brunswick Innova 44R, Eppendorf SE, Hamburg, Germany) at 250 rpm. Protein expression was induced at OD600 ~ 0.6 by adding isopropyl β-d-1-thiogalactopyranoside (IPTG) to a final concentration of 0.1 mM, and cultivation was continued at 13 °C for 36 h. Cells were harvested by centrifugation (H1750R, Cence, Changsha, China) at 7000× *g* for 10 min at 4 °C and stored at − 20 °C for later use.

The protein purification protocol is described in detail in [42]. Briefly, the cells were lysed in Lys buffer (20 mM Tris, pH 8.0, 250 mM NaCl, 0.2 mM phenylmethylsulfonyl fluoride, 1 mM EDTA, 10 mM β-mercaptoethanol) by sonication (Bandelin SONOPULS HD 2200, Bandelin, Berlin, Germany). The lysate was clarified by centrifugation (H1750R, Cence, Changsha, China) at 14,000× *g* for 30 min at 4 °C, and the membrane fraction was harvested by centrifugation at 150,000× *g* for 1 h at 4 °C. The precipitate was resuspended in IMAC buffer (20 mM Tris, pH 8.0, 250 mM NaCl, 10 mM Imidazole, 10 mM β-mercaptoethanol, 0.5 % lauryl sarcosine) and purified by immobilized metal affinity chromatography using Ni-Sepharose HP resin (GE HealthCare, Chicago, IL, USA). The fractions containing the target protein were combined, and the protein was further purified by size-exclusion chromatography (SEC) in SEC buffer (10 mM Tris, pH 8.0, 100 mM NaCl, 10 mM β-mercaptoethanol, 0.5 % lauryl sarcosine). To prepare liposomes with p75-TM-ICD, the lipid was added to the protein sample, and detergent was removed using BioBeads SM2 resin (BioRad, Hercules, CA, USA) [42]. Detergent removal was monitored by ^1^H-NMR.

### 2.3. Lipodisc Preparation

To prepare the lipodisc samples, stock mixtures of lipid MLVs and SMA-EA or SMA-tau in water were used. Lipid and SMA stocks contained 20 mM phosphate buffer (pH 8) and 5% D_2_O. The final concentration of SMA was 1% w/w, and the final concentration of lipid depended on the desired mass ratio of lipid to SMA, q = mass_lipid_/mass_SMA_. Samples were incubated in a thermomixer (Thermomixer comfort, Eppendorf SE, Germany, Hamburg) at 30 °C overnight.

To prepare the DMPC/DMPG/SMA samples, lipids were mixed in chloroform and dried using a centrifugal vacuum concentrator (Savant SPD1010, Thermo Scientific, Waltham, MA, USA). Next, 100 mM PBS buffer (pH 8) was added to the lipids, and the resulting mixture was vortexed and supplied with the SMA stock. The final concentration of SMA was 0.75% (w/w), and the final concentration of lipids was 1.5% (w/w) (individual lipid concentrations depended on the desired DMPC/DMPG ratio). Samples were incubated in the thermomixer at 30 °C overnight.

To prepare SMALPs containing an *E. Coli* membrane fraction with p75-TM-ICD, 20 mM phosphate buffer (pH 8) was added to the membrane fraction, the resulting mixture was vortexed, and the SMA stock was added afterwards. The final concentration of *E. coli* lipids was 0.55% (w/w), and the final concentration of SMA depended on the desired q ratio. Samples were incubated in the thermomixer at 303 K overnight and centrifuged (Centrifuge 5417 R, Eppendorf SE, Germany, Hamburg) at 25,000× *g* at 4 °C for 120 min.

### 2.4. Size-Exclusion Chromatography

Size-exclusion chromatography (SEC) experiments were performed on a Superdex 200 Increase 10/300 GL column (GE HealthCare, Chicago, IL, USA). An amount of 500 µL of the 2% w/w SMA-EA/tau sample or 500 µL of the 2% w/w DMPC/SMA-EA/tau sample with a lipid/SMA mass ratio of q = 1 was injected into the column. The working buffer was PBS pH 7.5. The flow rate was 0.5 mL/min.

### 2.5. Dynamic Light Scattering

Dynamic light scattering (DLS) experiments were performed on a DynaPro instrument (Wyatt Technology, Santa Barbara, CA, USA) using a 12 μL cuvette. Data were recorded in 25 scans of 5 s each. DLS data were processed using Wyatt Dynamics 6.7.7.9 software. Each measurement was repeated 3 times, and the mean size distribution was calculated.

### 2.6. NMR Spectroscopy

^1^H diffusion spectra for the DMPC/SMA-EA and DMPC/SMA-tau samples were acquired on a Bruker Avance 700 MHz spectrometer equipped with a room temperature triple resonance probe (Bruker, Billerica, MA, USA) using a double-stimulated echo pulse sequence with the suppression of convection effects and the solvent signal [43]. The diffusion delay was set to 400 ms.

^31^P NMR spectra were recorded on a Bruker Avance III 600 MHz spectrometer equipped with a broadband double resonance probe (Bruker, Billerica, MA, USA). A single pulse experiment was used, with a π/6 pulse of 4 μs, a relaxation delay of 3 s, and an acquisition time of 2.7 s, and with broadband WALTZ16 proton decoupling at a field strength of 1.42 kHz. A pre-acquisition delay of 10 min was used before each experiment to ensure the equilibrium state of the system.

^1^H-^15^N-HSQC spectra of p75-TM-ICD were recorded on Bruker Avance III 800 MHz and Bruker Avance 700 MHz spectrometers equipped with cryogenic triple-resonance probes (Bruker, Billerica, MA, USA).

### 2.7. NMR Data Processing

Diffusion NMR spectra were processed using Wolfram Mathematica software. The diffusion of DMPC was measured using signals at 0.9 and 1.3 ppm. The dependencies of DMPC/SMA particle radii on the q ratio were approximated by the ideal bicelle model with a cylindrical rim:(1)$$R\left(q\right)=r(1+\frac{q}{\lambda }+\frac{\sqrt{q(q+\lambda )}}{\lambda })$$
where *R* is the SMALP radius, *r* is the width of the rim, formed by SMA molecules, and *λ* is the ratio between the lipid and SMA densities [36]. 

Based on the ideal bicelle model approximation, the average amount of DMPC molecules in the NMR-observable particles was estimated using the following formula:(2)$$N_{lipids}=\frac{2\pi {\left(R-r\right)}^{2}}{S_{0}},$$
where *R* is the SMALP radius, *r* is the width of the rim, formed by SMA molecules, and *S*_0_ is the surface area of a single DMPC polar headgroup (was taken equal to 0.6 nm^2^ [44]).

To determine the chemical shifts, signals in the ^31^P spectra were deconvoluted using the Lorentzian lineshapes and nonlinear regression in Wolfram Mathematica software. The temperature dependence of the chemical shift was approximated by the following function:(3)$$\delta \left(T\right)=\left(a_{1}T+b_{1}\right)\frac{Exp\left[c\left(d-T\right)\right]}{1+Exp\left[c\left(d-T\right)\right]}+\left(a_{2}T+b_{2}\right)\frac{Exp\left[c\left(T-d\right)\right]}{1+Exp\left[c\left(T-d\right)\right]}$$
the inflection point of this function was calculated to determine the temperature of the phase transition.

## 3. Results

Our approach is based on three major techniques. First, by studying the dependence of particle size on the lipid/polymer ratio, q, we tested different models of the SMALP structure [36]. Second, we used ^31^P NMR spectroscopy to monitor the lipid phase transitions in the SMALPs [37]. Finally, using a model membrane protein, we analyzed the prospects of lipodiscs for solution NMR [42,45,46,47]. These three steps were taken to study the nanodiscs composed of two SMA-based polymers, modified with ethanolamine (SMA-EA) and taurine moieties (SMA-tau) (Figure 1). The ability of the SMA-EA copolymer to form lipodiscs of defined sizes, which can be controlled by adjusting the q ratio, makes it one of the most versatile SMA polymers [41]. The SMA-tau modification, a novel analog of SMA-EA in which the hydroxyl group is replaced with sulfonic acid, has not been proposed previously and is introduced in this study for the first time. We hypothesized that this modification could increase the solubility and monodispersity of SMA and SMA-based lipodiscs due to the double-negative charge on the modified maleic acid groups.

### 3.1. SMA-EA and SMA-Tau Form SMA–Lipid Particles

As previously reported, SMA forms particles with lipids [48]. Therefore, the first step of our work was to estimate the size distribution of lipodisc particles and to find out if the particles are suitable for solution NMR studies. We used size-exclusion chromatography (SEC) and compared the profiles of pure SMA-EA and SMA-tau (derivatives hereafter referred to as SMA) and their aggregates with DMPC lipids (Figure 2a). First, the elution volumes for the main peaks of SMA-EA and SMA-tau correspond to particles with a mass of about 40 kDa. At the same time, the SMA-EA profile is highly asymmetric and has a wide dispersion. This indicates an oligomerization of SMA-EA in solution, which is in agreement with previously reported data, where the formation of micellar species was observed [48]. Secondly, the profiles of the DMPC/SMA samples are shifted to the left and show a narrower, distinct peak in contrast to pure SMA. This indicates the formation of mixed massive DMPC/SMA particles with almost all SMA molecules being in the lipid-bound state. Based on the column calibration (Figure 2a), the molecular weight of the DMPC/SMA particles (1:1, w/w) exceeds 150 kDa, which corresponds to the expected size of lipodiscs (~5 nm in radius).

We analyzed the size of DMPC/SMA particles using dynamic light scattering (DLS) and NMR spectroscopy (Figure 2b–d). The results are consistent with the data provided by SEC. The peak fractions of the DMPC/SMA-EA and DMPC/SMA-tau mixtures consist of particles with average radii of 5.22 ± 0.05 and 4.33 ± 0.04 nm measured by DLS and of 4.36 ± 0.03 and 4.12 ± 0.06 nm measured by NMR. At the same time, it should be noted that there are some discrepancies in the values obtained by the different methods. We believe that this is explained by the fact that in NMR, larger particles contribute less to the signal, whereas in DLS, larger particles contribute more. Thus, there is most likely a particle size distribution in the sample, with the center lying between the DLS and NMR values.

In conclusion, SMA-EA and SMA-tau are able to interact with lipids and form particles with a size of ~10 nm, which corresponds to conventional lipid–protein nanodiscs used in solution NMR spectroscopy [49].

### 3.2. SMA-EA and SMA-tau Nanoparticles Reveal Bicelle-Like Behavior

To control the size and evaluate the morphology of the particles at typical working temperatures, we examined the sizes of DMPC/SMA-EA and DMPC/SMA-tau particles as a function of the DMPC/SMA w/w ratio, q, using the NMR diffusion measurements as described previously [43]. Since NMR diffusion measurements are less demanding in terms of sample preparation, and are more accurate and reproducible than the DLS, we continued our study with NMR as the main approach. The dependence of the average SMALP hydrodynamic radii on q for the DMPC/SMA-EA and DMPC/SMA-tau samples at 303 K and 313 K is shown in Figure 3.

For both DMPC/SMA-EA and DMPC/SMA-tau, the average lipodisc radius grows with increasing lipid content in the samples, similar to phospholipid bicelles. With this in mind, we applied the ideal bicelle model to describe the obtained dependence [36,50]. The ideal bicelle model is an equation that relates the size of the particle to the composition of the mixture, assuming the complete separation of the components within the particle, and can generally be applied to any type of discoidal aggregate. For SMA mixtures, we used the model that assumes a cylindrical shape of the lipodisc rim, which has two parameters: rim thickness and the ratio of the volumes that are occupied by the disc and rim molecules [36]. According to this analysis, SMA-EA forms a rim as thick as 2.8 nm at 303 K and 2.5 nm at 313 K, and SMA-tau forms rims of 3.0 and 2.7 nm at 303/313 K, respectively (Appendix A).

Based on the ideal bicelle model approximation, the calculated average number of lipid molecules per SMA-EA particle increased from 9.0 to 96.3 in the q range of 0.2 to 1.2 at 313 K. The average number of lipid molecules per SMA-tau particle increased from 3.3 to 50.7 at 313 K in the q range of 0.2 to 1.6 (Appendix A). Therefore, the DMPC/SMA-EA samples demonstrate a more rapid increase in lipid number compared to DMPC/SMA-tau, with a ratio of over twofold at q = 1.0 at both 303 K and 313 K, for instance. 

It is noteworthy that, in contrast to bicelles, the DMPC/SMA-EA samples exhibit pronounced temperature-dependent behavior. The SMA-EA–lipid particles are substantially larger at higher temperatures, which could be an undesired effect for the NMR applications. Like bicelles, DMPC/SMA-EA particles deviate from the ideal bicelle model starting from a specific SMA/lipid ratio, a phenomenon not observed in DMPC/SMA-tau mixtures [36]. 

We also tested how pH, ionic strength, and volume fraction of SMALPs affect lipodisc radius. Changes in pH moderately affected the DMPC/SMA-EA particles. An increase in pH from 7.0 to 8.0 resulted in an expansion of the particles, reaching 14% at 303 K and 20% at 313 K (Appendix A). These changes can be attributed to the differing charges of the polymers present in the rim. Interestingly, the radii of the DMPC/SMA-tau particles remained almost unaffected. The addition of NaCl up to 100 mM had no pronounced effect on the DMPC/SMA-EA particle radii (Appendix A). These particles showed moderate growth (~8% at 303 K and ~12% at 313 K), with an increase in volume fraction from 1.5% to 3% (Appendix A). 

To summarize, the radii of DMPC/SMA-EA and DMPC/SMA-tau particles behave similarly to bicelles and can be analyzed using the ideal bicelle model [36]. This model reveals that the SMA molecules form a thick rim, and the fraction of the volume occupied by the SMA in the SMALP particle is greater than the fraction occupied by the lipid for the q values considered. One might even wonder whether the lipid packing in SMALPs is disturbed and whether the properties of the lipids are similar to those in lipid bilayer membranes.

### 3.3. Phase Transitions in SMA Nanoparticles Reveal the Presence of Lipid Bilayer

Phase transition is a fundamental property of a lipid bilayer. Therefore, the study of such transitions in lipid–SMA samples would allow us to conclude whether lipodiscs mimic the properties of lipid bilayers. For this purpose, we analyzed the temperature-dependent behavior in the ^31^P NMR spectra of the lipid–SMA samples. The presence of inflection in the temperature–chemical shift dependence plots indicates the process of lipid phase transition [37].

Temperature dependencies of the DMPC ^31^P chemical shift for the DMPC/SMA-EA samples with q 0.3, 0.6, and 1.0 and DMPC/SMA-tau samples with q 0.6, 1.0, and 1.6 are shown in Figure 4a,b. No inflection is observed in the plots until the q value is higher than 0.6 in the case of the DMPC/SMA-EA samples and 1.0 in the case of the DMPC/SMA-tau samples, and this inflection is more pronounced at higher q values, which allows us to determine the critical temperatures of the phase transitions (Appendix A, Figure 4d, Appendix A). Furthermore, the DMPC/SMA plots are similar to those observed for the DMPC/DHPC bicelles and lipid–protein nanodiscs [37] (Figure 4c).

The next step of the work was to study lipodiscs with alternative lipid compositions in order to simulate various properties of the lipid bilayer, such as thickness or charge (Figure 5). For this purpose, ^31^P NMR spectra were obtained for POPC/SMA-EA (q = 1.5), DPPC/SMA-EA (q = 1.5), DMPC/DMPG/SMA-EA, and DMPC/DMPG/SMA-tau (q = 2) at various DMPC/DMPG ratios (60/40, 70/30, and 80/20).

First, no phase transition was observed in the POPC/SMA-EA particles (Appendix A). This is consistent with the fact that the phase transition temperature of POPC is below the water melting point [52]. For the DPPC/SMA-EA samples, the phase transition was observed at 311.3 ± 1.2 K, which is also in agreement with the higher phase transition temperature reported for DPPC bilayers [53] (Appendix A, Appendix A).

For the DMPG-containing samples, changes in the DMPC/DMPG ratio as well as the choice of SMA modification did not reveal a significant effect on the phase transition temperature (Figure 5 and Appendix A). 

Thus, phase transitions reveal the presence of a lipid bilayer in the particles with a sufficiently high lipid/SMA ratio. The phase behavior of different lipids in SMALPs is similar to the behavior of these lipids in bilayer membranes; therefore, the properties of the lipids are likely to be retained to a certain extent in the lipid–SMA particles under investigation.

### 3.4. Effect of the SMA Rim on Membrane Protein

So far, we have demonstrated that the tested SMALPs exhibit bicelle-like behavior and contain a patch of lipid bilayer that undergoes phase transitions [36,37,39], which is observed in the q range of 0.6–1.4 for SMA-EA and 1.0–1.6 for SMA-tau.

As a final step, we investigated the applicability of SMA-EA and SMA-tau lipodiscs as membrane mimetics for membrane proteins. As a model protein, we selected a fragment of the human p75 receptor, comprising the transmembrane and intracellular domains (p75-TM-ICD). NMR spectra of this protein have been previously obtained in various mimetics: bicelles, LPNs, and liposomes [42,45,46,47]. Furthermore, this protein is trafficked to the *E. coli* membrane during expression, allowing us to test the ability of SMA to extract the protein from the membrane.

We prepared protein-containing SMALP samples by extracting p75-TM-ICD from either POPC liposomes or the *E. coli* membrane fraction (MF), recorded the ^1^H-^15^N-HSQC NMR spectra (Figure 6 and Appendix A), and compared them with spectra of the same protein obtained directly in liposomes or the membrane fraction.

In the initial spectra of p75-TM-ICD (Figure 6a,c), the signals of the mobile ICD domain are observed. The TMD signals are broadened and invisible because TMD is embedded in the large lipid particles (liposomes or membrane vesicles). The addition of SMA-EA to the protein in POPC at q = 1.3 resulted in a dramatic decrease in signal intensity (Figure 6b). The addition of SMA-EA to the protein in MF at q = 2.0 left the intensity unchanged (Figure 6c,d), but further addition of SMA to q = 1.5 also resulted in a significant decrease in intensity (Figure 6e). Unlike SMA-EA, the addition of SMA-tau resulted in signal loss at both q = 2.5 and q = 1.5 (Appendix A). 

The observed effect of intensity decrease upon SMA addition could be caused either by protein precipitation (since the effect is proportional to the amount of SMA added) or by the interaction between the ICD domain and SMA molecules. To test the first hypothesis, we compared the intensities of the signal corresponding to the amino group of the C-terminal residue before and after SMA addition. The intensities of this signal remained weakly affected (Appendix A), ruling out protein precipitation. Therefore, we assume that there are transient interactions between the ICD domain and SMA molecules, leading to a decrease in ICD mobility. 

To quantitatively assess the immobilization of the ICD, we calculated the average decrease in NMR cross-peak intensity of the protein amide groups. The values were normalized to the intensity of the C-terminal amide group signal and plotted in Figure 6f. In the case of the MF/SMA-EA sample, at q = 2.0, almost no intensity decrease is observed after SMA addition compared to q = 1.5, where signal intensities decreased twofold. In contrast, in the case of MF/SMA-tau, there is no significant difference in spectral quality for either q value (Appendix A).

To summarize, we show here that both SMA-EA and SMA-tau can efficiently extract MPs from liposomes and cell membranes; however, the quality of the NMR spectra could be strongly affected by the protein–SMA interaction, as previously shown for SMA derivatives [34].

## 4. Discussion

The search for membrane mimetics that sufficiently reproduce the native membrane environment of MPs and at the same time are compatible with NMR, X-ray crystallography, and CryoEM remains a relevant task of modern structural biology. Bilayer-containing discoidal particles, bicelles and LPNs, have proven to be a suitable choice in several cases [1,2,3,4,5,6,7,8,9,10,11,12], but they have a number of disadvantages and limitations, including non-specific affinity, e.g., for water-soluble proteins. Therefore, the expansion of the existing set of available membrane mimetics is essential for the structural biology of membrane proteins. Analysis of the physical and chemical properties characteristic of hydrophobic particles formed in membrane mimetic solutions is also important to understand the limits of their applicability and to rationally select the mimetic for a particular membrane protein. In the current work, we applied the methodology previously developed in our group to investigate small isotropic bicelles [36,37,39] to study the structure and physical properties of SMALPs, a recently introduced and actively developed type of membrane mimetic [13,14,26,27,54,55,56].

First, we proposed the novel SMA derivative, SMA-tau. The polymer carries a negatively charged taurine moiety, an addition to the carboxyl group of SMA, which makes it a unique dianion in the set of available SMA variants [27,28,29]. We investigated the formation and properties of lipid–SMA-EA and lipid–SMA-tau membrane-mimicking particles. These SMALPs form particles with molecular weights greater than 150 kDa, which is similar to conventional lipid–protein nanodiscs. Using diffusion and ^31^P NMR in solution, we have shown that the lipid–SMA particles exhibit bicelle-like behavior, and their structural parameters can be obtained from the ideal bicelle model [36]. Both SMAs form a rim that is at least two times thicker than the detergent rim of commonly used bicelles and three times thicker than the rim thickness of lipid–protein nanodiscs [57]. At the same time, the data obtained here differ from the values presented in previous works [58,59,60,61,62]. According to the measurements obtained by SANS/SAXS methods, the rim thickness varies in the range of 0.6–1.8 nm depending on the SMA modification. This discrepancy can be explained by one of the following possibilities: (1) the SMA derivatives obtained here form a special rim that is different from the previously studied SMA; (2) the packing is highly dependent on the q parameter and the ideal bicelle model we used does not describe this system correctly; (3) the double-cylinder model used to process the SAXS data does not allow obtaining the correct rim thickness values due to the smooth transition between the disk and rim areas of the particles. Considering that the SMA-EA modification has been characterized previously, the problem is most likely explained by the inaccuracy of one of the models. Nevertheless, the studied SMALPs contain a lipid bilayer patch that can undergo phase transitions at temperatures close to those of lipid bilayer membranes [51,52,53]. 

Next, we employed SMA-EA and SMA-tau to extract a fragment of the p75 receptor from POPC liposomes and the membrane fraction of *E. coli* cells. The extraction was successful to some extent; however, we observed undesired interactions between the water-soluble domains of p75 and the SMA rim of the lipodiscs, and similar effects were previously found for the modified SMA lipodiscs and cytochrome p450 [34]. These interactions were stronger in the case of SMA-tau, making the protein almost unobservable in the NMR spectra. We believe that there are some non-specific interactions between the soluble domain of p75 and SMA. Higher values of q correspond to larger bilayer sizes, resulting in a reduced probability of the contact between the p75 intracellular domain and the SMA rim. This explains why we observed a decrease in signal intensity as the q value decreased. Thus, we can provide additional evidence that the properties of the SMA rim are extremely important for the solubilization of the membrane proteins under investigation. In this regard, the introduction of SMA-tau could be a valuable development. For the particular case of p75NTR, SMA-tau does not provide a significant improvement over SMA-EA; however, the situation may be different for the other proteins. Therefore, the introduction of SMA-tau, with its unique physical and chemical properties, expands the existing palette of SMA derivatives and may be useful for structural studies of other membrane proteins with different molecular surface properties. In addition, the overall smaller size of SMA-tau–lipid particles compared to SMA-EA particles formed at the same q is potentially advantageous for NMR applications.

SMALPs have numerous advantages over micelles, bicelles, and LPNs. They allow avoiding the use of detergents that can unfold the target protein [14,15,16,17,18,19,20,21,22]. Moreover, SMA polymers can extract the protein directly from lipid membranes, preserving the native lipid environment. The properties of SMA polymers can be altered by various covalent modifications to adapt to different pH values. SMALPs have demonstrated their applicability in a number of studies of membrane proteins by CryoEM [30,31,32]. However, as with other membrane mimetics, SMALPs have their drawbacks and limitations, especially when it comes to their application for the needs of solution NMR spectroscopy. As we have found here, the size of SMA-EA SMALPs depends on the temperature, which is an undesirable phenomenon, considering that the object size is an important parameter that defines the quality of NMR spectra. Furthermore, for the low q values (<0.6 for SMA-EA and <1.0 for SMA-tau), where the relatively small particles are formed, we did not observe any phase transitions, indicating that the environment provided by these particles is far from native. Conversely, at high q values (>1 for SMA-EA and SMA-tau), excessively large particles were found, which are not applicable for solution NMR. Furthermore, the experiments with the p75 protein revealed that the ICD domain can interact non-specifically with the rim. Thus, although these SMA polymers have all the properties of beneficial membrane mimetics, their use in solution NMR is limited and varies with the system under investigation. Screening experiments are necessary for each particular protein.

## Figures and Tables

**Figure 1 polymers-16-03009-f001:**
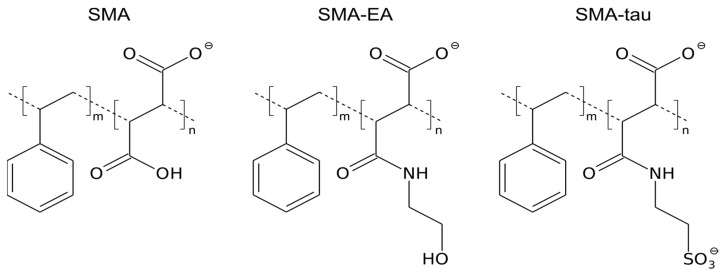
The chemical structure of SMA and its modifications SMA-EA and SMA-tau.

**Figure 2 polymers-16-03009-f002:**
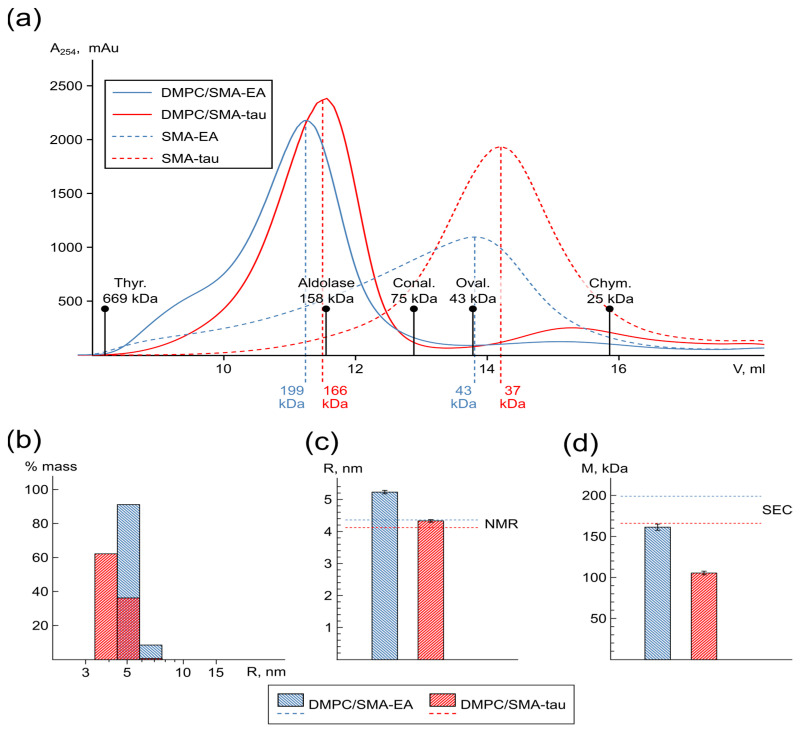
Size of SMA and SMA–lipid particles. (**a**) SEC profiles of SMA-EA (dashed blue line), DMPC/SMA-EA (blue line), SMA-tau (dashed red line), and DMPC/SMA-tau (red line) samples. (**b**) Distribution of radii for DMPC/SMA-EA (blue bars) and DMPC/SMA-tau (red bars) samples at 303 K obtained by DLS. The DMPC/SMA ratio in all cases was 1:1 (w:w). (**c**) Average radii of DMPC/SMA-EA (blue bar) and DMPC/SMA-tau (red bar) particles measured by DLS and NMR (blue and red dashed lines, respectively). (**d**) Average mass of DMPC/SMA-EA (blue bar) and DMPC/SMA-tau (red bar) measured by DLS compared to SEC data (blue and red dashed lines, respectively).

**Figure 3 polymers-16-03009-f003:**
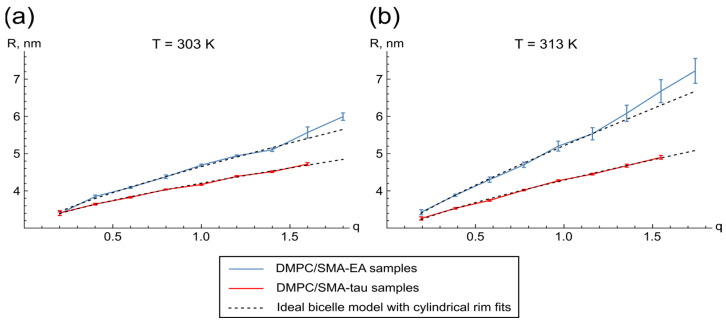
Dependencies of the lipid–SMA particle radii on q (**a**) obtained at 303 K and (**b**) at 313 K. Blue and red lines correspond to the data for DMPC/SMA-EA and DMPC/SMA-tau samples, respectively. Dashed lines correspond to their approximation by the ideal bicelle model with a cylindrical rim. Sample buffer contained 20 mM NaPi, pH 8.

**Figure 4 polymers-16-03009-f004:**
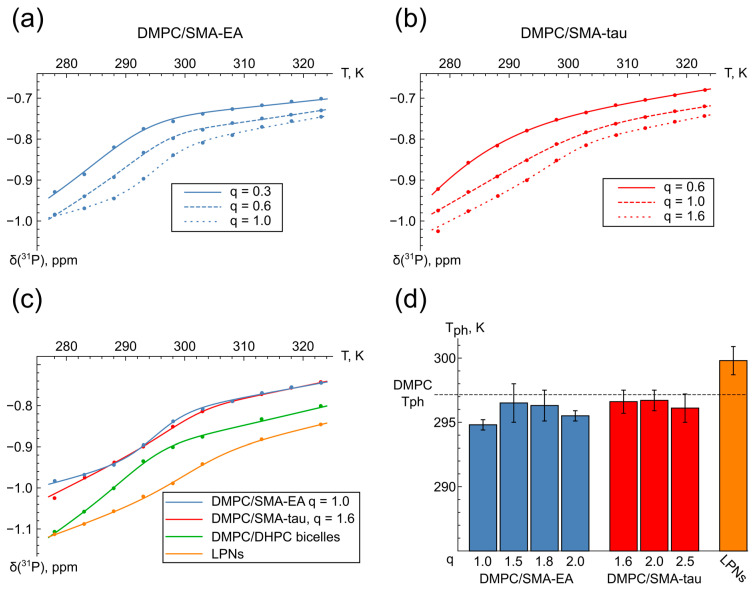
Analysis of chemical shifts in ^31^P spectra at different temperatures. Temperature dependence of chemical shifts for (**a**) DMPC/SMA-EA samples and (**b**) for DMPC/SMA-tau samples (**c**). Temperature dependence of chemical shifts for DMPC/SMA samples with DMPC/DHPC bicelles and lipid-protein nanodiscs (LPNs); (**d**) Phase transition temperatures for DMPC/SMA-EA, DMPC/SMA-tau, and LPNs compared with available data for DMPC in a lipid bilayer (dashed line) [51].

**Figure 5 polymers-16-03009-f005:**
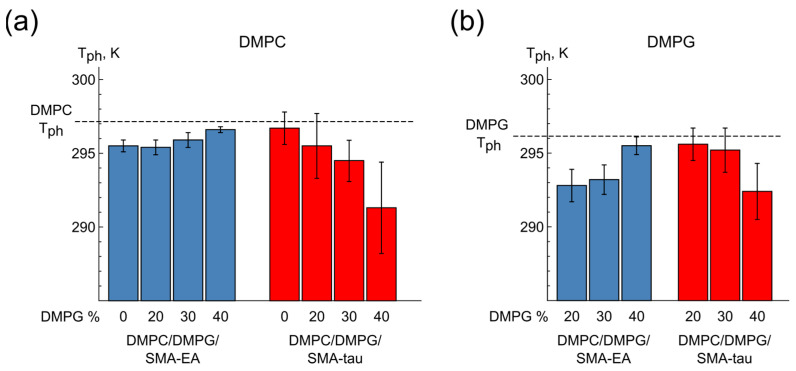
Phase transition temperatures of (**a**) DMPC and (**b**) DMPG in DMPC/DMPG/SMA-EA and DMPC/DMPG/SMA-tau samples compared to literature data for DMPC and DMPG bilayers (dashed lines) [51].

**Figure 6 polymers-16-03009-f006:**
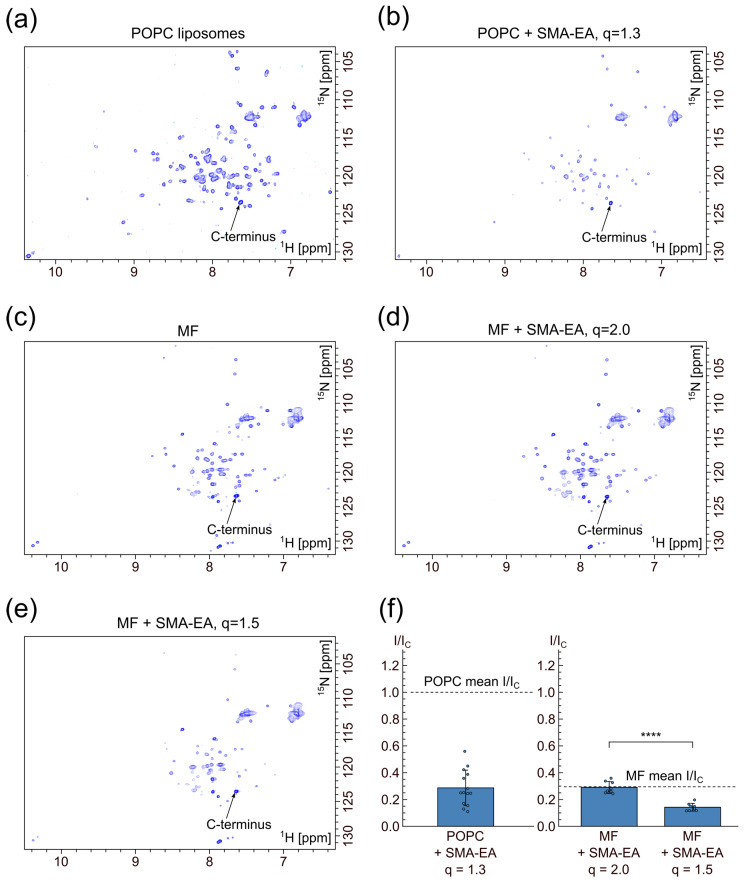
^1^H-^15^N-HSQC spectra of p75-TM-ICD in different membrane mimetics: (**a**) in POPC liposomes; (**b**) in POPC/SMA-EA at q = 1.3; (**c**) in *E. coli* membrane fraction (MF); (**d**) in MF/SMA-EA at q = 2.0; (**e**) in MF/SMA-EA at q = 1.5; (**f**) relative change in cross-peak intensities in the ^1^H-^15^N-HSQC spectra of p75-TM-ICD after the SMA-EA addition.

## Data Availability

All the data are available on request to the authors.

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
