# Peer review of "On the Properties of Styrene–Maleic Acid Copolymer–Lipid Nanoparticles: A Solution NMR Perspective"

_polymers, 2024, doi:10.3390/polym16213009_

Round 1
Reviewer 1 Report
Comments and Suggestions for Authors
Vladislav V. Motov et al. investigated the application of two novel SMA molecules for membrane protein extraction and NMR spectroscopy. This study contributes to the growing work on SMA in studying membrane proteins, particularly in structural analysis.
However, I have some reservations about the manuscript, detailed point by point below:
- In this study, they measured the size of SMALPs using DLS and NMR. It would be better if they also performed electron microscopy for size measurements, if accessible.
- In line 210, they stated the diameter is about 10 nm, as most studies suggest. However, the measurement in this paper shows a diameter of around 5 nm. It would be better to adjust this statement to reflect their data.
- In line 339, they mentioned that the ICD interacts transiently with SMA molecules. It is also likely to interact with lipids, and this interaction could be more significant. As the SMA extracts the native lipid from the cell membrane.
- There is a typo in line 15: the abbreviation for styrene and maleic acid should be SMA, and SMALPs should stand for styrene and maleic acid lipid proteins.
Author Response
Reviewer #1:
- In this study, they measured the size of SMALPs using DLS and NMR. It would be better if they also performed electron microscopy for size measurements, if accessible.
- Unfortunately, we do not have access to an electron microscope. Thus we cannot perform this experiment in the short-term.
- In line 210, they stated the diameter is about 10 nm, as most studies suggest. However, the measurement in this paper shows a diameter of around 5 nm. It would be better to adjust this statement to reflect their data.
- We measure diffusion coefficients by NMR and then calculate the radius of the particles. Therefore, in our NMR results, we refer to the radius of the particles rather than the diameter. A radius of 5 nm corresponds to a diameter of 10 nm, aligning perfectly with the literature and our SEC and DLS data. To avoid any confusion, we replaced the diameter with the radius in the revised manuscript.
- In line 339, they mentioned that the ICD interacts transiently with SMA molecules. It is also likely to interact with lipids, and this interaction could be more significant. As the SMA extracts the native lipid from the cell membrane.
- Indeed, the protein could interact with some components of the E. coli membrane. Moreover, the spectrum of the ICD domain in the membrane fraction has a worse quality compared to liposomes or bicelles/nanodiscs, due to the interaction with some components. At the same time, we have a control experiment with POPC liposomes. As you can see in Figure 6A, the spectrum of the ICD domain is well resolved, with many distinct narrow signals. However, after the addition of SMA (Figure 6B), the quality of the spectra significantly worsens. Moreover, this effect is proportional to the amount of SMA added. Taking into account that no additional components were added to the system, we conclude that SMA interacts with the protein. We modified the text slightly to make our considerations clear.
- There is a typo in line 15: the abbreviation for styrene and maleic acid should be SMA, and SMALPs should stand for styrene and maleic acid lipid proteins.
- Corrected.

Reviewer 2 Report
Comments and Suggestions for Authors
I have several questions/comments that need to be addressed before this manuscript can be considered for publication:
-
In Figure 1, panels C and D present the DLS data compared to NMR and SEC, and there is an obvious discrepancy between these techniques. The authors should provide an explanation for this discrepancy.
-
The authors have used NMR diffusion to measure particle sizes instead of DLS. The rationale for this choice should be explicitly provided in the manuscript to clarify why this approach was preferred.
-
The authors state: "A slight change was observed for DMPC/DMPG/SMA-tau particles with increasing DMPG content; however, the high error value does not allow confirmation of the reliability of this trend." I recommend rephrasing this sentence—if the trend is not statistically reliable, then its existence should not be implied.
Author Response
Reviewer #2:
- In Figure 1, panels C and D present the DLS data compared to NMR and SEC, and there is an obvious discrepancy between these techniques. The authors should provide an explanation for this discrepancy.
- Indeed, there are discrepancies in the values of different methods. However, it should be noted that in NMR, particles with a larger size make a smaller contribution to the signal, since their movements are slower, which leads to broadening of the signals. Contrary, in the DLS the particles with a larger size make a greater contribution to the signal. Thus, most likely, there is a particle size distribution in the sample, with the center lying between the values measured by DLS and by NMR. We added this clarification to the text.
- The authors have used NMR diffusion to measure particle sizes instead of DLS. The rationale for this choice should be explicitly provided in the manuscript to clarify why this approach was preferred.
- Indeed, DLS can be used for such studies, however, this method is more demanding in terms of sample preparation. First, as one could see in Figure 2A, the peaks on the SEC profile are quite broad, which indicates the heterogeneity of the sample. DLS models describe such mixtures very poorly and usually, each repetition provides different values. In addition, large particles contribute much stronger to the DLS signal, compared to the smaller ones. Thus, prior purification by the SEC is required for DLS measurements. In the case of NMR, such purification is not necessary, because the large oligomeric complexes are not visible in the spectra. Second, one of our aims is to check the suitability of these SMALPs for NMR applications. Thus, it seemed logical to us to use this method. We added the rationale to the manuscript at the beginning of Section 3.2.
- The authors state: "A slight change was observed for DMPC/DMPG/SMA-tau particles with increasing DMPG content; however, the high error value does not allow confirmation of the reliability of this trend." I recommend rephrasing this sentence—if the trend is not statistically reliable, then its existence should not be implied.
- We agree with the reviewer and have removed this phrase.